# Development of a Novel Stress and Immune Gene Panel for the Australasian Snapper (*Chrysophrys auratus*)

**DOI:** 10.3390/genes15111390

**Published:** 2024-10-29

**Authors:** Kerry L. Bentley-Hewitt, Christina K. Flammensbeck, Ross N. Crowhurst, Duncan I. Hedderley, Maren Wellenreuther

**Affiliations:** 1The New Zealand Institute for Plant and Food Research Limited, Private Bag 11600, Palmerston North 4442, New Zealand; 2The New Zealand Institute for Plant and Food Research Limited, Nelson Research Centre, Box 5114, Port Nelson, Nelson 7043, New Zealandmaren.wellenreuther@plantandfood.co.nz (M.W.); 3The School of Biological Sciences, The University of Auckland, Private Bag 92019, Auckland 1142, New Zealand; 4The New Zealand Institute for Plant and Food Research Limited, Private Bag 92169, Auckland 1142, New Zealand

**Keywords:** aquaculture species, snapper, gene expression, stress resistance

## Abstract

Background: Snapper (*Chrysophrys auratus*) is a commercially, recreationally and culturally important teleost species in New Zealand and has been selected as a potential new species for aquaculture. Selective breeding to enhance stress tolerance, survival and growth are major breeding targets, yet research into snapper immune and stress responses has been limited. Methods: We explored a set of candidate genes in the fin, head kidney and liver tissues of 50 individuals by exposing 20 fish to increasing temperature (up to 31 °C) and 20 fish to decreasing temperature (down to 7 °C) for up to 37 h. Of these, we analysed 10 temperature-sensitive and 10 temperature-tolerant fish, along with 10 fish kept at 18 °C (acclimation temperature) as a control group. Results: Expression analyses of candidate stress genes in the three tissue types via NanoString Technologies, Inc., Seattle, WA, USA. showed that 20 out of 25 genes significantly changed in each experiment, demonstrating the significant impact of temperature on stress and immune responses. We further document that 10 key gene biomarkers can be used to predict genotypes that are tolerant to extreme temperatures. Conclusions: Taken together, our novel NanoString method can be used to monitor stress in snapper rapidly, and applications of this tool in this and potentially closely related teleost species can provide insights into stress resilience of wild stocks and inform the selection of grow-out locations for aquaculture.

## 1. Introduction

Stress responses involve the neural, endocrine and immune systems and can be acute (increasing innate responses) or chronic (immune-suppressive and increasing the risk of infection) [1]. Temperature has a significant impact on the stress and immune responses in virtually all animal species [2,3], particularly in ectotherm animals like fishes [1,4]. The majority of teleost species are ectothermic, and immune responses can depend on seasonal changes, such as temperature increases causing enhanced immune responses and temperature decreases suppressing responsiveness to antigens [5]. Innate immunity is of fundamental importance in preventing pathogen entry and consists of physical (e.g., skin and mucus), humoral (e.g., complement proteins) and cellular components (e.g., macrophages) [6]. Research into the immune systems of fish has been limited by reagents for classical immunology studies [7]. However, with the availability of a species transcriptome, gene expression can be used to investigate potential biomarkers that are indicative of an animal’s health status [2,8]. Biomarkers are crucial tools for the assessment of environmental changes [9,10]. Fish show conservation of most immune-related genes with vertebrates, but investigations into functional immunology remain scarce [6].

Antioxidant enzymes are often used in toxicology as sensitive biomarkers, e.g., superoxide dismutase (*sod*) and catalase (*cat*), which constitute the first line of defence against reactive oxygen species (*ros*). Besides antioxidant enzymes, another group of genes involved in the immune response is induced under oxidative stress conditions because of the relationships between an imbalance in antioxidant properties and susceptibility to diseases, which includes the antimicrobial peptide hepcidin (*hamp*). The great sensitivity to a wide range of stress conditions makes the gene expression of these antioxidant enzymes and immune-related genes suitable for use as biomarkers in teleost fishes [9].

The Australasian snapper, *C. auratus*, tāmure, (hereafter referred to as snapper) is a species that was selected as a candidate for aquaculture in New Zealand, and a selective breeding program was started in 2016. Snapper is a valuable commercial and recreational fish species located around the coasts of Australia and New Zealand [11,12]. It is a demersal marine finfish species that can reach a total length of 120 cm and weights of around 20 kg [12]; it is closely related to *Pagrus*/*Chrysophrys major* (hereafter referred to as red seabream) (Temminck & Schlegel 1843) [13], a major aquaculture species in Japan [14,15,16]. In New Zealand, it mostly inhabits the continental shelf at depths of up to 200 m, but primarily above 70 m in the upper section of the epipelagial zone, on rocky, muddy or sandy seabeds. Predominantly, snapper is distributed around the coast of the North Island and the northern coast of the South Island, with decreasing abundance further south caused by lower water temperatures, which are a relevant parameter for the reproduction cycle [17,18,19]. Snapper has a wide distribution across New Zealand and Australia, inhabiting nearly all inshore environments down to depths of 200 m [12,20]. In New Zealand, snapper is predominantly found in the northern half of the North Island and the northern part of the South Island [12], occasionally occurring in the southern South Island [21], where temperatures can go well below 10 °C. Like many other Sparidae species, they experience a broad range of natural temperatures [19,22,23,24]. For instance, their range extends as far north as Mackay, where sea surface temperatures (SSTs) can reach up to 30 °C [25]. Moreover, experimental work has repeatedly shown that snapper can adjust effectively to both warm and cold temperature treatments, exhibiting minimal mortality even after months of exposure [23,26]. Herein, we aimed to exceed the species’ thermal tolerance limits by exposing snapper to extreme temperatures outside of the seasonal temperature range since climate change may expose snapper to these extremes in the future. Breeding of snapper in New Zealand started around 20 years ago at The New Zealand Institute for Plant and Food Research Limited (Plant & Food Research) in Nelson, but selective breeding using genomic information was only initiated in 2016 with the aim of improving the growth rate and other traits of economic interest [27]. Prior to the selective breeding research program, only a handful of genetic markers were available for this species [28,29]. Our group has been developing diverse resources for snapper to facilitate rapid breeding progress in this species, including a genome, linkage map, transcriptome and genome-wide sequence information on pedigreed snapper [23,27,30,31]. Research into snapper immune and stress responses has been limited so far, yet this is an important trait for the domestication process and the selective breeding of species. Stress and immunity in closely related species, such as red seabream and gilthead seabream (*Sparus aurata*) [32,33,34,35], have been studied. Both species are already successfully developed for use in commercial aquaculture in Japan and the Mediterranean, respectively. These studies focus on a range of factors that affect stress and immune responses, such as infection, pollution and dietary interventions [36,37,38,39].

In this study, we aim to identify and develop novel biomarkers associated with the thermotolerance of Australasian snapper using NanoString Technologies, Inc. (Seattle, WA, USA). This is the first time such an approach has been applied to this species, making it a novel contribution to the field. By focusing on genes responsive to different temperature settings, we enhance our understanding of the physiological and molecular mechanisms underpinning the snapper’s resilience to thermal stress. This focus on biomarkers facilitates the selection of breeding stock with superior thermal tolerance, contributing to a deeper understanding of the snapper’s adaptive capacity in fluctuating environments. Here, we explore a set of genes identified as potentially being involved in the stress and immune responses of snapper exposed to an extreme temperature challenge as follows: either a heating (up to 31 °C) or cooling treatment (down to 7 °C) for up to 37 h, with temperature-sensitive fish (those that started to show signs of distress) and temperature-tolerant fish (those that survived the temperature extreme) being sampled along with control fish (kept at the acclimation temperature of 18 °C). We (1) first established a list of 97 candidates from the literature and then (2) identified key gene biomarkers in snapper that can be used to predict genotypes that are tolerant to extreme temperature changes in the following tissues: fin, head kidney and liver tissues. We discuss these findings and compare them with other studies to highlight novel results and outline future applications based on the use of the developed gene panel.

## 2. Materials and Methods

### 2.1. Experimental Design

All work conducted in this study was approved by the Animal Ethics Committee at the University of Auckland, New Zealand, under ethics approval reference number 002169.

Temperature experiments were conducted at Plant & Food Research’s Finfish facility in Nelson. The F_3_ study population was derived from a genomically selected core F_2_ broodstock population consisting of 63 individuals. Breeding took place over five consecutive days at the end of November 2019. Fertilized eggs were collected daily and placed in a 5000-litre self-cleaning tank for initial rearing. The experiment was designed to study acute stress response of juvenile snapper (~4–6 months old) exposed to increasing (heating treatment, up to 31 °C) and decreasing temperatures (cooling treatment, down to 7 °C) for up to 37 h.

In this acute temperature experiment, a total of 600 juvenile snapper were divided into the two main treatment groups as follows: warm and cold. The warm treatment involved three tanks with 100 fish each, where the temperature was initially set at 18 °C and gradually increased to 22 °C over 24 h, followed by an extreme heating protocol raising the temperature by approximately 1 °C per hour for 9 h to reach 31 °C. The cold treatment also involved three tanks with 100 fish each, starting at 18 °C and decreasing to 14 °C over 24 h, followed by extreme cooling at approximately 1 °C per hour for 7 h to reach 7 °C. Temperature, dissolved oxygen and other water quality parameters were monitored twice daily using a handheld YSI meter, with continuous temperature logging in each tank to ensure stable conditions throughout the experiment. Before the start of the experiment, 10 individuals were sampled as the control group, which was kept under the acclimation temperature (18 °C) and did not experience the temperature challenge.

Fish were continuously monitored throughout the experiment. Upon reaching the target temperature, the experiment was concluded as soon as 10 individuals exhibited signs of stress. For heat shock, these signs included agitated swimming and jumping behaviour, while for cold stress, the indicators included loss of equilibrium and sinking to the bottom of the tank. Those individuals were classified as temperature-sensitive and were sampled along with 10 temperature-tolerant individuals that were not in distress at the end of the experiment or showed the least signs of stress. Fish in the cooling treatment showed signs of distress before the 33 h timepoint; therefore, the fish that were last to show signs of distress were the cold-tolerant group. Fish in the heating treatment took longer than 37 h to show any signs of stress; therefore, the most stressed were the heat-sensitive group. For this study, a subset of samples was taken from one cooling and one heating replicate. The head kidney, liver and a fin sample were taken from a total of 50 individuals representing control *(n* = 10), temperature-sensitive (*n* = 20, 10 per treatment group) and temperature-tolerant individuals (*n* = 20, 10 per treatment group) (see Table 1 for phenotypic data on the different sample groups). The experimental design is shown in Figure 1.

### 2.2. Gene Target Sequence Design

Candidate genes were selected from a literature review of studies investigating at immune- and stress-related genes, mainly in red and gilthead seabream, and whether they could potentially be identified in the snapper transcriptome. We searched the following databases in April 2020: Web of Science core, current content connect™, CABI, FSTA, MEDLINE, Russian Science Citation Index, SciElO Citation Index (collectively Web of Science all) and Google Scholar (limited to first 5 pages over the previous 2 years). We used the keywords “*Pagrus auratus*” OR “*C. auratus*” OR “*Pagrus major*” OR “*S. aurata*” OR “*Tamure sparus*”) AND immun* refined to review papers, “*P. auratus*” OR “*C. auratus*” AND immun* AND (gene* OR “T cell” OR “pentraxin” OR “cytokines” OR “TNF” OR “IFN” OR “CXC” OR “iNOS” OR “COX”) and “*P. major* “ OR “*S. aurata*” OR “*T. Sparus*” AND (immun* OR cytokines) AND TITLE: gene* for Web of Science all searches. For Google Scholar searches, we used “*P. major*” OR “*S. aurata*” OR “*T. sparus*”) AND (immun* OR cytokines) AND TITLE: gene*. A summary of our search strategy is in Appendix A.

A list of 97 putative candidate gene sequences was identified within the NCBI Gilthead seabream (*S. aurata*) annotation release 100 (www.ncbi.nlm.nih.gov/genome/annotation_euk/Sparus_aurata/100/) URL assessed on 22 August 2019, and the coding sequences (CDS) for these were downloaded from NCBI. The 97 gilthead seabream CDS sequences were then aligned to the whole-genome assembly of snapper [32] using gmap (“--cross-species --format=gff3_gene --ordered --tolerant --gff3-fasta-annotation = 1 --min-trimmed-coverage = 0.85 --min-identity = 0.85”). The general feature format/general transfer format (GFF/GTF) utility gffread (https://github.com/gpertea/gffread) (version 0.12.1) was then used to extract the CDS and predicted peptide sequences from the snapper whole-genome sequence using the command line options “-i 50000 --merge -K -D --adj-stop”. This extraction yielded 73 CDS sequences, of which 71 CDS sequences, with a minimum length of 150 bases, were assessed for completeness.

The 71 sequences were used to design probes for a PlexSet assay by NanoString Technologies, Inc., and from these, a final set of 48 targets was selected. In several cases, the same gene annotation was given with a different isoform, e.g., the following five mucin 18-like isoforms: X, X2, X3, X4, and X5. In these instances, NanoString managed to design primers to bind to all isoforms, resulting in a list of 60 genes. Three sequences were deemed to be either truncated at their 5 prime terminus, truncated at their 3 prime terminus or a fragment truncated at the 5 prime and 3 prime ends relative to the original snapper query sequence; these were removed. Finally, the list was reduced to 48 genes by selecting those thought more likely to be modified by the temperature treatments, based on information in the current literature. Genes with “-like” in their name were discussed as if they were the gene. Annotation for the snapper and gilthead seabream genome is still limited, so the results must be interpreted with some caution.

In summary, candidate genes were selected from a literature review of studies investigating immune- and stress-related genes, mainly in red and gilthead seabream, and whether they could potentially be identified in the snapper transcriptome. The final set of gene targets and their potential link to fish stress and immune responses are summarized in Table 2.

### 2.3. RNA Extraction

Snap-frozen fin, head kidney and liver tissue samples (approximately 15–25 mg) were placed on ice and homogenized in 600 µL RA1 lysis buffer with 6 µL β-mercaptoethanol (NucleoSpin RNA, Mini kit, cat. no. 740955.50, MACHEREY-NAGEL GmbH & Co. KG, Düren, Germany). The samples were then filtrated using a NucleoSpin^®^ Filter and centrifuged at 11,000× *g* for 1 min at room temperature. An equal volume of 70% molecular-grade ethanol was added to the filtrate and transferred to a NucleoSpin^®^ Mini column (NucleoSpin RNA, Mini kit, cat. no. 740955.50, MACHEREY-NAGEL GmbH & Co. KG, Germany). RNA was extracted as per the manufacturer’s protocol into 60 µL of tris-ethylenediaminetetraacetic acid buffer for gene expression of 48 genes (Table 3) by the Counter Analysis System (NanoString Technologies, Inc.). RNA quantity was assessed by a Qubit™ RNA BR (Broad-Range) Assay Kit as per the manufacturer’s instructions (Invitrogen™, cat. no. Q10211, Waltham, MA, USA) and a Qubit^®^ 2.0 Fluorometer (Invitrogen™, USA).

### 2.4. Gene Expression Analysis—NanoString Counter Analysis System

RNA samples (64 ng fin, 21 ng head kidney and 45 ng liver) were analysed using nCounter Plexset reagents (NanoString, Seattle, WA, USA) following the manufacturer’s instructions. Target sequences were designed by NanoString Technologies, Inc. and ordered from Integrated DNA Technologies, Inc., Coralville, IA, USA (Table 3). Briefly, multiplexed probes are designed with two sequence-specific probes for each gene of interest. The capture probe is coupled to biotin as an affinity tag. The second probe, the reporter probe, is coupled to a colour-coded tag. Each target molecule of interest is identified by the unique colour code generated by the ordered fluorescent tags on the reporter probe. The level of expression is measured by counting the number of codes for each mRNA using digital imaging. This allows the analysis of multiple genes from the same sample (multiplexing) using a customized set of probes with distinct bar codes called a CodeSet [67]. The samples were prepared as previously described in Bentley-Hewitt et al. (2016) [68]. Initially, the samples were spiked with six different internal control probes at concentrations ranging from 128 fM to 0.125 fM in 4-fold dilution steps. Eight positive controls were used to determine the linearity of the assays and were used for normalization. Eight negative probes were used to control for the carryover of reporter probes, as no RNA target was included for these probes. A calibration sample containing a pool of each snapper tissue was run in eight lanes of the assay. To pass the assay calibration, at least 50 counts of each gene had to be present in each of the eight lanes. The RNA samples were incubated for 22 h at 65 °C in a hybridization buffer containing the CodeSet, which consisted of reporter and capture probes and, together with the target RNA, formed a tripartite complex. After hybridization, the complex was bound by its biotin-labelled capture probe on a streptavidin-coated glass slide and was stretched within an electric field. The hybridized samples were processed using the robotic Prep Station (High Sensitivity Protocol, 3 h per 12-sample cartridge), and data acquisition was performed by using the GEN2 Digital Analyzer, with the “Max” Field of View setting (555 images per sample; 5 h scan per cartridge). Raw counts were normalized using the positive controls, and target genes were normalized to the following internal reference genes: 40S ribosomal protein S18 (*rps18*), 60S ribosomal protein L8 (*rpl8*) and elongation factor 1-α (*ef1a*), using nSolver™ 4.0 analysis software (NanoString Technologies, Inc.).

### 2.5. Statistical Analysis

Analysis of variance (ANOVA) of the logs of gene expression counts was used to summarize the data from the fins, head kidney and liver. Fish was fitted as a random effect, and treatment, tissue type and their interaction as fixed effects. Fisher’s least significant differences were also calculated between the log-transformed expressions for different genes to compare treatments. To adjust for multiple testing, the significance level was set to 0.00033 (=0.05/(6*25)). The analyses were carried out using Genstat 20th edition (VSN International, Hemel Hempstead, UK).

## 3. Results

### 3.1. Gene Expression

The expression of genes varied considerably, and to avoid overloading the NanoString cartridges, the concentration of RNA added was low, with our previous study using 800 and 1450 ng of organ tissue [69]. This resulted in many genes on the panel having low counts, which could not be calibrated by the assay’s nSolver™ 2.0 software, as each gene requires 50 counts in the eight lanes containing the calibration sample. These genes had to be removed from the analysis, leaving 25 target genes and three reference genes. One of the reference genes *gapdh* was removed because of its variability in the samples, and it was treated as a target gene. The final target genes were normalized to three reference genes that were stable and passed calibration.

### 3.2. Short-Term Gene Expression Experiment

Mean gene counts are shown in Table 4 for snapper exposed to heating (up to 31 °C) and cooling temperature (down to 7 °C) for up to 37 h, along with fish kept at the acclimation temperature (18 °C), including fish that were temperature-sensitive (those that started to show signs of distress) and temperature-tolerant (those that survived the temperature extreme). The mean counts ranged from 0 to 130,275.

There were 20 significant changes in genes when comparing heating- and cooling-treated fish to the control group and when comparing temperature-tolerant snapper to temperature-sensitive snapper for each treatment.

The temperature effects on gene expression varied across tissue types and the stressor type (e.g., heating versus cooling). In general, there were only differences in gene expression between tolerant and sensitive snapper with the heating treatment. For example, *gsr*, *hsp70*, *igf2* and *ptgs2* were expressed at significantly different levels when comparing heat-sensitive with heat-tolerant fish in at least one tissue type. In contrast, the expression did not differ between cold-sensitive and cold-tolerant fish in any tissue (Figure 2, Figure 3 and Figure 4 and Table 4). The most consistent gene expressed across tissues was *cry1-like*. This gene resulted in a significantly lower expression with heating treatments than the control group and significantly higher expression with cooling treatments than the control group.

### 3.3. Genes Affecting Tolerance to Temperature Extremes

An additional statistical analysis (ANOVA) was conducted to determine which genes had a higher tolerance to temperature extremes, taking into account all tissue types. More genes appeared to influence tolerance to extreme heating conditions. These genes were *muc18-like* (*p* = 0.025), *cry1-like* (*p* = 0.013), *gsr* (*p* ≤ 0.001), *hsp70* (*p* = 0.005), *igf2* (*p* = 0.035), *ucp2-like* (*p* = 0.011), *prdx1* (*p* = 0.041), *ptgs2* (*p* ≤ 0.001) and *sod* (*p* = 0.027), whilst only *gsr* (*p* = 0.046), *hsp70* (*p* = 0.007) and *hsp90* (*p* = 0.007) appeared to influence tolerance to cold temperature extremes.

## 4. Discussion

Here, we developed a set of 25 candidate stress genes for snapper fin, liver and head kidney tissues to detect signs of stress rapidly (Aim 1). We explored these candidate genes relative to three reference genes to determine if potential stress and immune genes were linked to the tolerance of fish to extreme temperature changes (Aim 2).

Our literature search revealed that stress responses affect numerous genes and are dependent on the stressor and the species of fish (summarized in Table 2). For example, overcrowding stress in gilthead seabream modulated the expression of *actb* (decreased expression), *c3* (increase expression) and *prdx1* (decreased expression) in skin tissue [41], whereas post-sediment exposure in gilthead seabream, the expression of *gsr* and *il1b* decreased in liver and skin tissue, *hamp* in head kidney and *csfr1* in both skin and head kidney [9]. Our previous research showed that *actb*, *cry1*, *gpx2*, *hsp70*, *hsp90*, *ucp2* and *soc3* expression in snapper white muscle tissue correlated with temperature changes when exposed to high and low temperatures [23]. Another study exposing grass carp to high and low temperatures showed increases in antioxidant enzyme (*gst*, *gpx1* and *gsr*) expression in the liver at high temperatures but not low temperatures [49]. In summary, research into stress response genes was limited in snapper; however, some studies investigated stress responses in phylogenetically close fish species, although the stressors varied among research studies.

When analysing the tissues from the temperature experiment, we found that nine genes influenced tolerance to extreme heating conditions compared with just three to extreme cooling conditions. *muc18-like* was expressed significantly higher in heat-tolerant fish fin compared with heat-sensitive fish, presumably because this gene leads to the upregulation of mucin production (a first-line defence response against pathogens), suggesting that heat-tolerant snapper have heightened immune defences. *cry1-like* was shown to significantly affect survival at extreme heating conditions. This is in line with previous work on this species, where *cry1* was shown to be modulated in response to high and low temperatures [23]. *gsr* expression increased in the liver and head kidney of heat-tolerant snapper compared with heat-sensitive snapper. This gene produces an antioxidant enzyme that aids in the first line of defence against reactive oxygen species (ROS) [43], indicating that heat-tolerant fish are more adapted to deal with ROS stress, which may aid survival in temperature extremes. Conversely, a down-regulation of *gsr* was observed in the liver of cold-tolerant snapper, showing that increases in this gene may not always aid survival and could be dependent on the type of stress applied (e.g., heating versus cooling). *hsp90* mean expression was often lower in the tolerant snapper at both temperature extremes, with the exception being in the head kidney of heat-tolerant snapper. Previous research has shown that this gene increases at temperature extremes in the white muscle tissue of snapper and supposedly increases protection from oxidative stress [23]; therefore, it is unclear why we observed a decrease in tolerant fish, unless these fish did not experience as much oxidative stress via other mechanisms. *igf2* was down-regulated in the liver of heat-tolerant fish compared with heat-sensitive fish, but it is not clear if this may aid survival. Additionally, *ucp2-like* decreased in the fins of heat-tolerant fish. This gene is involved in the maintenance of oxidative processes inside cells [37]. This may indicate that heat-tolerant snapper are not as exposed to oxidative stress, which may aid their survival. However, *prdx1* increased in the head kidney of heat-tolerant snapper; this gene plays a role in the maintenance of oxidative and anti-oxidative processes in cells [37]. *ptgs2* plays a key role in inflammation in fish [61]. It increased in the liver of heat-tolerant fish, which may suggest a higher expression of this gene aids survival at hotter temperature extremes. Lastly, *sod* decreased in the liver of heat-tolerant fish, which, owing to its role in the first line of defence against ROS, supports the theory that heat-tolerant snapper experience less ROS stress. In summary, there appear to be some genes that offer a survival advantage to extreme heat (e.g., *ptgs2*, *prdx1* and *muc18-like*); however, in general, it may be that these fish are better adapted to avoid oxidative-related stress than their sensitive counterparts. The general down-regulation of genes involved in stress protection in cold-tolerant snapper may also be indicative of the same lack of oxidative stress.

One limitation of this study is that annotations for the snapper and gilthead seabream genomes are still limited, so the results must be interpreted with some caution. Work is currently underway to use an improved genome assembly for snapper [70] to add improved gene annotations, and this may support future efforts to develop an improved panel. Also, genes with -like in their name have been discussed as if they were the gene. Additionally, this study used captive fish, which may differ from wild fish, and this could mean that stress responses were limited or exacerbated [71,72]. Identifying genetic traits that increase tolerance to temperature changes carries significant applied value as this knowledge can be used to selectively breed snapper for these traits to improve resilience to temperature fluctuations and ultimately to make this a more commercially viable aquaculture species. This is urgently needed in New Zealand because of the increasing impacts of climate change on the aquaculture sector [73,74,75].

To summarize, overall, we present a key set of 25 stress- and immune-related genes in various tissues, along with three reference genes, to explore their changes when snappers were exposed to increasing and decreasing temperatures. Twenty of the twenty-five genes did change significantly, demonstrating the significant impact of water temperature on stress and immune responses. We document 10 key gene biomarkers that could be used to predict genotypes that are more tolerant to extreme temperature changes. We recommend that future work should focus on improving the annotation of the snapper genome and ensuring that the genes identified here confer to protein production changes in snapper tissues. In addition, more studies on a wider range of snapper from various locations around New Zealand, plus individuals from other species, are needed to determine if the genes identified could serve as a general tool for monitoring fish health under thermal stress conditions. Taken together, our novel NanoString tool can be used to monitor stress in snapper rapidly and can possibly be transferred to closely related teleost fishes. Applications of this tool could be used to provide insights into the stress resilience of wild populations and help with the selection of grow-out locations for aquaculture.

## Figures and Tables

**Figure 1 genes-15-01390-f001:**
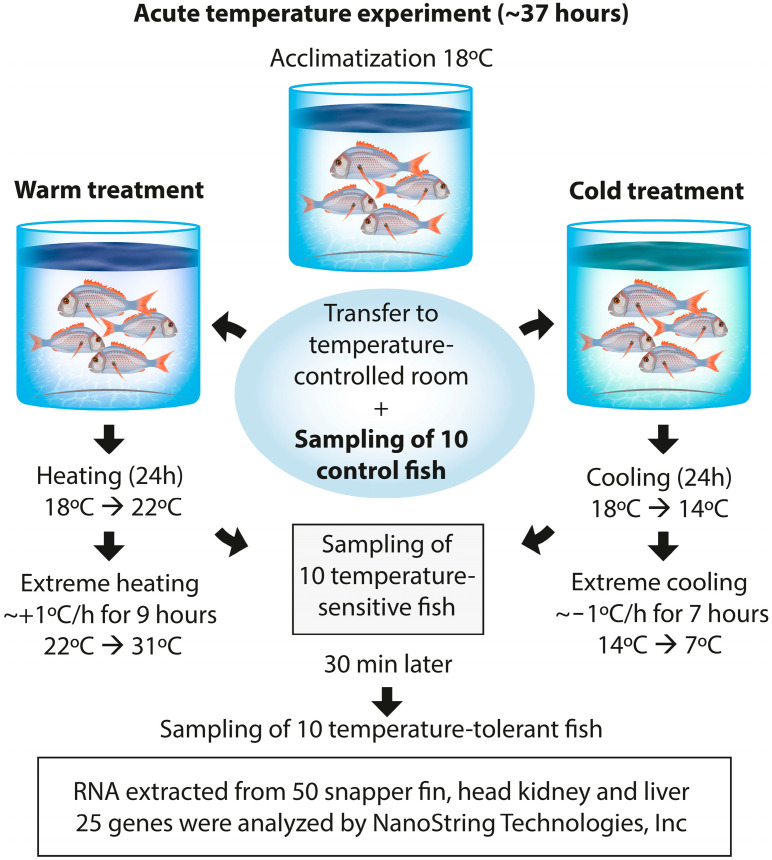
Schematic overview of the experimental design.

**Figure 2 genes-15-01390-f002:**
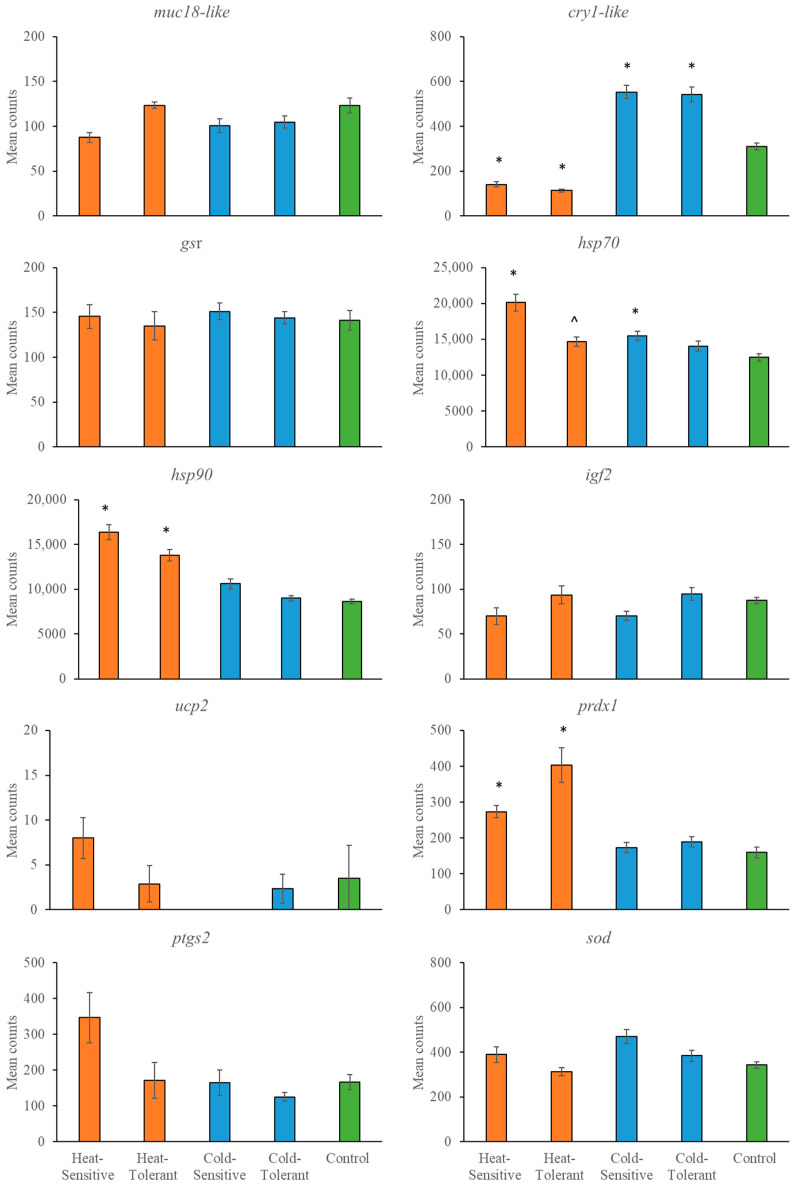
Mean gene counts (*n* = 10) from fin tissue are shown with standard errors for heat-sensitive, heat-tolerant, cold-sensitive, cold-tolerant and control fish. A significant difference (*p* < 0.00033) is indicated by * for differences compared with the relevant control, whilst ^ indicates a difference between the heat-sensitive and heat-tolerant groups.

**Figure 3 genes-15-01390-f003:**
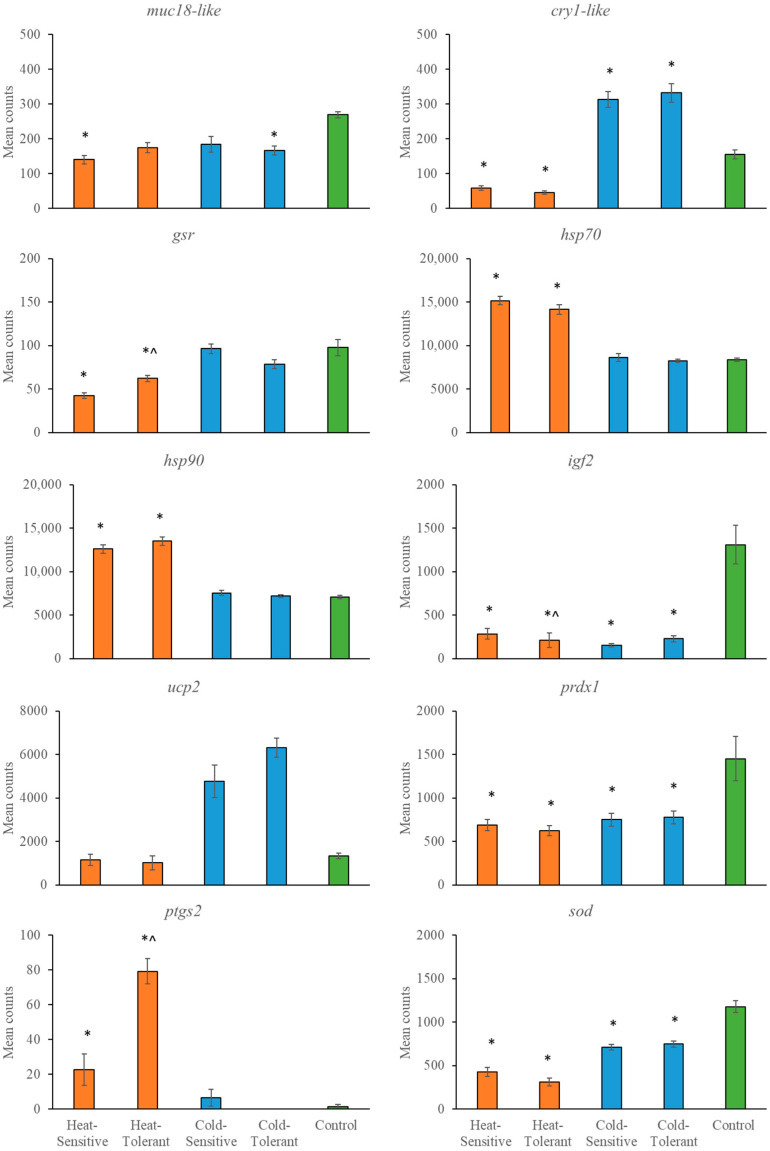
Mean gene counts (*n* = 10) from liver tissue are shown with standard errors for heat-sensitive, heat-tolerant, cold-sensitive, cold-tolerant and control fish. A significant difference (*p* < 0.00033) is indicated by * for differences compared with the relevant control, whilst ^ indicates a difference between the heat-sensitive and heat-tolerant groups.

**Figure 4 genes-15-01390-f004:**
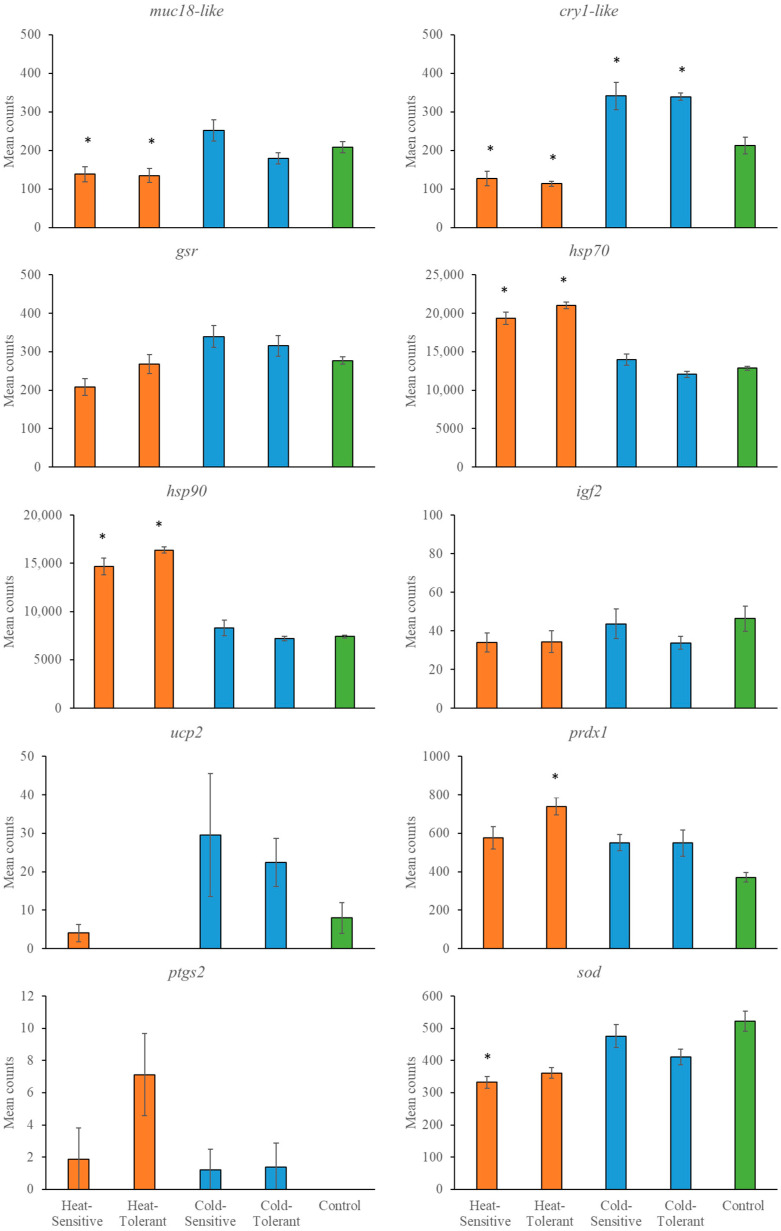
Mean gene counts (*n* = 10) from head kidney tissue are shown with standard errors for heat-sensitive, heat-tolerant, cold-sensitive, cold-tolerant and control fish. A significant difference (*p* < 0.00033) is indicated by * for differences compared with the relevant control.

**Table 1 genes-15-01390-t001:** Length (mm, mean ± SE) and weight (g, mean ± SE) measurements of 50 juvenile snapper exposed to temperature challenge and sampled for gene expression analysis.

Treatment Group	Sample Size (*n*)	Fork Length [mm]	Weight [g]
Control	10	111.7 ± 3	29.8 ± 3
Cooling: Temperature-sensitive	10	107.1 ± 4	26.2 ± 3
Cooling: Temperature-tolerant	10	116.0 ± 4	33.7 ± 4
Heating: Temperature-sensitive	10	108.1 ± 3	25.1 ± 2
Heating: Temperature-tolerant	10	113.6 ± 3	30.9 ± 2

**Table 2 genes-15-01390-t002:** Genes and their potential function and/or relation to fish stress and immune responses.

Gene	Function/Role in Stress or Immune Responses
Actin cytoplasmic 1 (*actb*)	ACTB is important for phagocytosis [40] and is decreased in gilthead seabream skin owing to over-crowding stress [41]. *actb* is correlated with temperature changes in white muscle tissue of snapper exposed to high and low temperatures [23].
Alkaline phosphatase tissue-nonspecific isozyme isoform X1 (*alpl*)	Involved in intestinal defence [42]. A plant-based diet decreased alkaline phosphatase in gilthead sea bream, which may reflect decreased innate immunity and be a reason for higher mortality [39].
Catalase isoform X2 (*cat*)	An antioxidant enzyme used in toxicology as sensitive biomarkers and constitutes the first line of defence against reactive oxygen species (ROS) [43].
Cell surface glycoprotein MUC18-like isoform X (*muc18*)	*muc18* is the predominant mucin in the skin, gills and stomach in gilthead sea bream [44].
Complement C3-like A (*c3-like*)	Central component of complement is cleaved into C3b, the main effector molecule of the complement system, and C3a, a peptide mediator of inflammation [45]. Moringa leaf increased *c3* gene expression in gilthead sea bream post-hydrogen peroxide exposure [46]. *c3* increased in the skin of gilthead seabream after overcrowding stress [41].
Complement component C6 (*c6*)	Terminal complement protein, forms membrane attack complex and lysis certain pathogens and cells [45]. *c6* decreased after *Aeromonas hydrophila* infection in catfish skin [47].
Complement component C8 α chain (*c8a*)	Terminal complement protein, forms membrane attack complex and lysis certain pathogens and cells [45].
Cryptochrome-1-like (*cry1-like*)	Circadian gene modulated by high and low temperatures in snapper [23].
Glyceraldehyde-3-phosphate dehydrogenase (*gapdh*)	Involved in the breakdown of glucose for energy.
Glucocorticoid receptor-like (*gcr-like*)	Regulates the stress response and binds heat shock proteins, such as HSP70 [48].
Glutathione S-transferase A (*gsta*)	The glutathione-related antioxidant system is important for intercellular defence. *gst* increased in the liver at temperatures above 30 °C when grass carp were exposed for 28 days, but not with cold temperatures down to 15 °C [49].
Glutathione peroxidase 2 (*gpx2*)	*gpx2* is correlated with temperature changes in white muscle tissue of snapper exposed to high and low temperatures [23]. The glutathione-related antioxidant system is important for intercellular defence. *gpx1* increased in the liver at temperatures below 20 °C when grass carp were exposed for 28 days, whilst temperatures of 25 °C and above did not change its expression [49].
Glutathione reductase mitochondrial isoform X2 (*gsr*)	An antioxidant enzyme used in toxicology as a sensitive biomarker and constitutes the first line of defence against ROS [43]. *gsr* in the liver and skin tissue of gilthead seabream decreased post-sediment exposure [9]. The glutathione-related antioxidant system is important for intercellular defence. *gsr* increased in the liver at temperatures below 20 °C when grass carp were exposed for 28 days, whilst temperatures of 25 °C and above did not change its expression [49].
Heat shock cognate 70 kDa protein (*hsp70*)	Involved in cytoprotection, cell survival and immune response. It is inducible throughout inflammation and represents an effort to avoid apoptosis [50]. *hsp 90* and *70* increased at temperature extremes in snapper white muscle and offered protection from oxidative stress and apoptosis [23]. They facilitate a stress response by increasing binding of the steroid receptor [48].
Heat shock protein HSP 90-β isoform X2 (*hsp90ab1*)	Involved in cytoprotection, cell survival and immune response. It is inducible throughout inflammation and represents an effort to avoid apoptosis [50]. *hsp 90* and *70* increased at temperature extremes in snapper white muscle and offered protection from oxidative stress and apoptosis [23]. They facilitate a stress response by increasing binding of the steroid receptor [48]. Polyvinyl chloride microparticles decreased *hsp90* in the liver of gilthead sea bream [37]. *hsp90* increased after *A. hydrophila* infection in catfish skin [47].
Hepcidin (*hamp*)	An antimicrobial peptide [51]. *hamp* in gilthead seabream head kidney decreased post-sediment exposure [9].
Insulin-like growth factor I isoform X1 (*igf1*)	IGF1 stimulates natural killer cell function and macrophage production of ROS and cytokines [1]. IGF-1-can indicate higher growth [52].
Insulin-like growth factor II (*igf2*)	IGF2 potentially stimulates muscle growth more than IGF1 [53].
Interferon-induced GTP-binding protein Mx-like B (*mx-like*)	Viral-induced gene in red seabream [54].
Interleukin-1 β-like (*il1b-like*)	Stimulates proliferation and activation of macrophages and T cells and initiates an acute phase response [5]. The stress response is activated by TNFalpha, IL1B, IL-6 and IL-12 [1]. It activates target cells by binding IL1 receptors on cell surfaces and triggers inflammation to cope with infection [55]. Hydrogen peroxide increased *il1b* and *tnf-α* (pro-inflammatory cytokines) in gilthead seabream [46]. Intestinal *il1b* increased post-skin wounds in gilthead seabream [56]. *il1b* in gilthead seabream liver and skin decreased post-sediment exposure [9]. *il1b* decreased after *A. hydrophila* infection in catfish skin [47].
Interleukin-6 isoform X1 (*il6*)	The stress response is activated by IL6 [1].
Interleukin-10 (*il-10*)	In mammals stress hormones inhibit T helper lymphocyte type 1 (Th1) responses and cause a shift to T helper lymphocyte type 2 (Th2), including increases in IL10, and cold shock in mammals increases IL10 [1]. Heat stress decreased IL-10 in the liver of rabbits [57].
Interleukin-12 subunit β-like (*il12b*)	Induced in macrophages after bacterial and viral infections and is suggested to increase the cytolytic properties of T cells against viral-infected cells [5]. Secreted by antigen-presenting cells after the activation of pathogen and damage-associated molecular patterns. Induces natural killer cell interferon-γ production [58].
Interleukin-17D (*il17d*)	A pro-inflammatory cytokine [51]. *il17a* increased after *A. hydrophila* infection in catfish skin [47].
Interleukin-34 isoform X1 A (*il34*)	*il-34* expression is induced by lipopolysaccharide, poly(I:C), *il1b*, interferon-γ, phytohaemagglutinin and parasitic infection in rainbow trout [59].
Macrophage colony-stimulating factor 1 receptor (*csf1r*)	Intestinal *csf1r* increased post-skin wound in gilthead seabream [56]. *csf1r* in gilthead seabream head kidney decreased post-sediment exposure [9]. Infection with lymphocystis in gilthead seabream decreased *csf1r* expression in the skin and head kidney [60].
Mitochondrial uncoupling protein 2-like (*ucp2-like*)	The mitochondria are the first responder to various stressors. UCP is involved in the maintenance of oxidative and anti-oxidative processes inside cells and polyvinyl chloride microparticles increased *ucp1* in the liver of gilthead seabream [39]. *ucp2* correlated with temperature changes in white muscle tissue of snapper exposed to high and low temperatures [23].
Mucin-2-like isoform X1 (*muc2-like*)	Intestinal *muc2* increased post-skin wound in gilthead seabream [56].
Nuclear factor NF-kappa-B p100 subunit isoform X1 (*nfkb2*)	NFKB2 is a subunit of the NF-kappa-B transcription complex (NFKB), which plays a crucial role in biological processes such as inflammation, immunity, differentiation, cell growth, tumorigenesis and apoptosis.
Nuclear factor erythroid 2-related factor 2 (*nrf2*)	Involved in antioxidant regulation.
Peroxiredoxin-1 (*pdrx1*)	PRDX is involved in the maintenance of oxidative and anti-oxidative processes inside cells [37]. *pdrx1* correlated with temperature changes in white muscle tissue of snapper exposed to high and low temperatures [23]. *pdrx1* decreased in gilthead seabream skin post-overcrowding stress [41].
Peroxiredoxin-1-like (*pdrx1*-*like*)	PRDX is involved in the maintenance of oxidative and anti-oxidative processes inside cells [37]. *pdrx1* correlated with temperature changes in the white muscle tissue of snapper exposed to high and low temperatures [23].
Peroxiredoxin-5 mitochondrial (*pdrx5*)	PRDX is involved in the maintenance of oxidative and anti-oxidative processes inside cells. *pdrx5* decreased in the head kidney of gilthead seabream fed polyvinylchloride microparticles [37].
Prostaglandin G/H synthase 2 (*ptgs2*)	Is a prostaglandin synthesis enzyme that plays a key role in inflammation in fish [61].
Serotransferrin-like (*tf-like*)	Serotransferrin forms part of fish innate immunity and serves as an antimicrobial agent [62].
Stromal cell-derived factor 1 (*cxcl12*)	Recruits IL10-producing lymphocytes and macrophages [63]. *cxcl12b* increased after *A. hydrophila* infection in catfish [47].
Superoxide dismutase [Cu-Zn] (*sod1*)	An antioxidant enzyme used in toxicology as a sensitive biomarker, which constitutes the first line of defence against ROS [43]. *sod* increased after *A. hydrophila* infection in catfish skin [47].
Suppressor of cytokine signalling 3 (*soc3*)	Negative feedback to inhibit pro-inflammatory cytokine signalling [63]. *soc3* correlated with temperature changes in the white muscle tissue of snapper exposed to high and low temperatures [23].
T-cell surface glycoprotein CD8 α chain-like isoform X3 (*cd8a-like*)	A seabass developmental study investigated genes (*tcrb*, *cd8-α* and *cd4*) indicative of T cell function and found that *cd8-α* was detected at 40–50 days post-hatching and continued to increase until 92 days post-hatching [64,65].
Thioredoxin-dependent peroxide reductase mitochondrial (*prdx3*)	PRDX is involved in the maintenance of oxidative and anti-oxidative processes inside cells [37].
Toll-like receptor 2 isoform X1 (*tlr2*)	Innate immune receptors that detect infections in mammals. TLR2 detects lipopeptides in fish [66].
Toll-like receptor 3 (*tlr3*)	Innate immune receptors that detect infections. TLR3 has been shown to respond to double-stranded RNA (dsRNA) in mammals [66].
Toll-like receptor 5 (*tlr5*)	Innate immune receptors that detect infections in mammals [66]. Recognizes bacterial flagellin. *tlr5a* decreased after *A. hydrophila* infection in catfish [47].
Transforming growth factor β-1 proprotein-like isoform X1 (*tgfb1-like*)	In mammals, stress hormones inhibit Th1 responses and cause a shift to Th2, including increases in TGFB, and cold shock in mammals increases TGFB [1]. Mediates resistance to infection by controlling pathogen replication and cell proliferation. Regulates inducible nitric oxide synthase and therefore nitric oxide production [5]. *tgfb*-like increased after *A. hydrophila* infection in catfish [47].

**Table 3 genes-15-01390-t003:** Gene targets analysed by NanoString Technologies, Inc.

Gene Name	GenBank Accession Number	Target Region	Target Sequence	HUGO Gene
Reference genes				
Elongation factor 1-α (*ef1a*)	Ch_aur001.1	1152–1251	CAAGAAGCTTGAGGATGCTCCCAAGTTCGTCAAGTCTGGTGATGCCGCCATTGTCAAACTGCACCCACAGAAGCCCATGGTTGTGGAGCCCTTCTCCAGC	LOC115578802
60S ribosomal protein L8 (*rpl8*)	Ch_aur022.1	634–733	GGTGGTGGTAACCATCAGCATATTGGCAAACCCTCAACAATCAGAAGGGACGCACCTGCTGGTCGCAAGGTCGGTCTCATTGCTGCCCGTCGTACAGGCA	rpl8
40S ribosomal protein S18 (*rsp18*)	Ch_aur036.1	184–283	GAGGTTGAGCGTGTGGTGACCATCATGCAGAATCCTCGCCAGTACAAAATCCCAGACTGGTTCCTCAACAGGCAGAAGGACGTCAAGGACGGCAAATACA	LOC115577508
Target genes				
Actin cytoplasmic 1 (*actb*)	Ch_aur069.1	736–835	CAGGTCATCACCATCGGCAATGAGAGGTTCCGTTGCCCAGAGGCCCTCTTCCAGCCTTCCTTCCTCGGTATGGAGTCCTGCGGAATCCACGAGACCACCT	actb
Catalase isoform X2 (*cat*)	Ch_aur025.1	641–740	GCTACGGCTCTCACACCTTCAAACTGGTCAATGCCAATGGTGAGCGTTTCTACTGCAAGTTCCACTACAAGACTGATCAAGGAATAAAGAATCTGACAGT	cat
Cell surface glycoprotein MUC18-like isoform X (*muc18-like*)	Ch_aur003.1	412–511	ACTTACTTTGTTCCTGGAGGAACCAGGATGACTGAGACCAACCGTATTAACATCACTGTATACTACCCCTCCACCGCTGTAAGTGTTTGGGTGGAGTCAC	LOC115570482
Complement C3-like A (*c3-like*)	Ch_aur061.1	2871–2970	TCTGATTCTCAATGCACAGCAACCTGACGGCATGTTTAAAGAAGTTGGAACGGTCTCCCACGGGGAGATGATTGGCGATGTGCGCGGCGCAGATTCAGAT	LOC115582848
Complement component C8 α chain (*c8a*)	Ch_aur047.1	568–667	TGGAGGAAATTCAGCTATGACTCATTCTGTGAGAACCTGCACTACAATGAAGATGAGAAGAACTACAGGAAACCTTACAACTACCACACCTACCGTTTTG	c8a
Cryptochrome-1-like (*cry1-like*)	Ch_aur034.1	1655–1754	ACCAACAAACCAGCATCGGAACACACCAGCAAGGTTATCCAGGTACCAGTGCCGGTGTGATGTGTTACACTCAAGGCACACCACAGCAGATTCCTGGTTC	LOC115595869
Glutathione reductase mitochondrial isoform X2 (*gsr*)	Ch_aur019.1	236–335	TCAATGTTGGCTGTGTCCCTAAGAAGGTTATGTGGAATGCTGCAAGTCACGCCGAGTATCTCCATGATCACAATGATTATGGCTTCGACGTTGGAAATGT	gsr
Glutathione S-transferase A (*gsta*)	Ch_aur021.1	284–383	AACTGGCAATGATGTACCAGCGCATGTTTGAGGGTCTCTCACTCAACCAGAAAATGGCGGATGTCATCTACTACAACTGGAAGGTCCCAGAGGGAGAGAG	LOC115579480
Glyceraldehyde-3-phosphate dehydrogenase (*gapdh*)	Ch_aur068.1	303–402	CTTGAAGGGTGGTGCCAAGAGAGTCATCATCTCTGCACCCAGCGCCGACGCTCCCATGTTTGTCATGGGTGTCAACCATGAGAAGTACGACCATTCCCTC	gapdh
Heat shock cognate 70 kDa protein (*hsp70*)	Ch_aur056.1	1605–1704	GGTGTCTGCTAAGAATGGCCTGGAGTCGTATGCTTTCAACATGAAGTCTACTGTGGAGGATGAAAAACTTGCTGGCAAAATCAGTGATGACGACAAGCAG	LOC115594641
heat shock protein HSP 90-β isoform X2 (*hsp90ab1*)	Ch_aur007.1	514–613	GGAGCTGACATCTCCATGATTGGTCAGTTTGGTGTGGGTTTCTACTCTGCCTACCTTGTTGCTGAGAAGGTGGTCGTCATCACCAAACACAACGATGATG	hsp90ab1
Hepcidin (*hamp*)	Ch_aur008.1	101–200	AGGAGGCAGGGAGCAATGACACTCCAGTTGCGGCACATCAAGAAATGTCAATGGAATCGTGGATGATGCCGAGTCGCGTCAGGGAGAAGCGTCAGAGCCA	hamp
Insulin-like growth factor I isoform X1 (*igf1*)	Ch_aur033.1	188–287	GAGAGAGAGGCTTTTATTTCAGTAAACCTGGCTATGGCCCCAATGCACGGCGGTCACGTGGCATTGTGGACGAGTGCTGCTTCCAAAGCTGTGAGCTGCG	igf1
Insulin-like growth factor II (*igf2*)	Ch_aur050.1	161–260	CGCTGTGTGGGGGAGAGCTGGTGGATGCGCTGCAGTTTGTCTGCGAAGACAGAGGCTTCTATTTCAGTAGGCCAACCAGCAGGGGAAACAACCGGCGCCC	igf2
Mitochondrial uncoupling protein 2-like (*ucp2-like*)	Ch_aur038.1	539–638	TCACTAGAAATGCGCTTGTCAACTGCACAGAACTGGTTACATACGACCTGATCAAGGAGGCCATCCTCAAACACAACCTGTTGTCAGACAACCTGCCCTG	LOC115579854
Nuclear factor erythroid 2-related factor 2 (*nrf2*)	Ch_aur065.1	1360–1459	CAGAGGGCTAAGGCCCTCAAAATCCCTTTCACTGTAGACATGATTATCAATCTGCCTGTCGACGATTTCAATGAGCTGATGTCAAAGCACCGACTGAATG	nfe2l2
Peroxiredoxin-1 (*prdx1*)	Ch_aur042.1	162–261	CGAGATCATAGCTTTCAGTGACGCTGCTGACGATTTCAGGAAGATCGGCTGTGAGGTCATCGCCGCCTCTGTTGACTCACACTTCTCCCATTTCGCATGG	LOC115573364
Peroxiredoxin-1-like (*prdx-like*)	Ch_aur013.1	374–473	CATACAGGGGGCTGTTTGTGATTGACGACAAGGGCATCTTGAGGCAGATCACCATCAATGACTTGCCTGTGGGTCGCTCTGTGGATGAGACTCTGCGCCT	LOC115587998
Peroxiredoxin-5 mitochondrial (*prdx5*)	Ch_aur057.1	152–251	TGTCTATGGATCAGCTCTTCAAGGGGAAGAAGGGAGTCCTCTTTGCTGTACCTGGAGCCTTCACACCTGGATGTTCCAAGACTCACCTCCCAGGTTTTGT	prdx5
Prostaglandin G/H synthase 2 (*ptgs2*)	Ch_aur071.1	1187–1286	TCGTCTTCAACACGTCTGTAGTGACTGAGCACGGCATCAGCAACCTTGTGGAGTCGTTTTCCAAGCAGATCGCTGGACGGGTTGCCGGTGGCCGAAATGT	ptgs2
Serotransferrin-like (*tf-like*)	Ch_aur005.1	582–681	CGAGCCTTATTATGACTACGGTGGAGCCTTCCAATGTCTGGCAGACGACGCTGGTGATGTGGCCTTTGTGAAGCATCTCACTGTACCTGAGTCTGAAAAG	LOC115572354
Superoxide dismutase [Cu-Zn] (*sod1*)	Ch_aur059.1	100–199	GGAGAAATCTCGGGACTTACTCCTGGTGAGCATGGTTTCCATGTCCATGCATTTGGAGACAATACAAATGGGTGCATCAGTGCAGGCCCTCACTTCAATC	sod1
Suppressor of cytokine signalling 3 (*socs3*)	Ch_aur039.1	291–390	GCGCATCCAGTGTGACTCAAGCTCTTTTTTCCTGCAGACGGACCCTAAAAACGTTCAGTCTGTTCCTCACTTTGACTGCGTCCTCAAGCTGGTGCATTAC	socs3
Thioredoxin-dependent peroxide reductase mitochondrial (*prdx3*)	Ch_aur006.1	290–389	CCTTTGTGTGTCCAACAGAGATCATCTCATTCAGCGACAAGGCCAGTGAGTTCCACGACGTTAACTGTGAGGTGGTGGGTGTGTCGGTGGACTCTCACTT	prdx3
Transforming growth factor β-1 proprotein-like isoform X1 (*tgfb1-like*)	Ch_aur024.1	300–399	CAGTGCCATCAATTTTGAGGTCTCCGGGATCTCGAATAGTAGGGGAGACACACAAGGGTTTCAACAGGTGTCGCAGCAATACCCGTACATCCTGACCATG	LOC115575711
Genes that did not pass calibration				
Alkaline phosphatase tissue-nonspecific isozyme isoform X1 (*alpl*)	Ch_aur035.1	661–760	GGCTGCAAGGATATCGCCAGACAACTCTTTGAAAATATTCCCAACATTGATGTGATTATGGGTGGAGGAAGGAAGTATATGTTCCCTAAAAACAAGTCGG	alpl
Complement component C6 (*c6*)	Ch_aur018.1	2413–2512	CTCTGTATCCTGAACGTAGACCTCGGCGTCACCGTGCCGATGTCCCTCTGCTCCTTCCACGTCGGGCTTTGCCACAATGATCCGCTCTTCTATGTCAGCG	c6
Glucocorticoid receptor-like (*gcr-like*)	Ch_aur049.1	669–768	GGACGTTGGCTCAGAGAGGGACATGAAGTCTGCTGTGGTTGAAAGCATTAACGGCAGTGGGGCAGTCTTTGTTGCTCTTAATGGCAGTAATATGACAAGT	LOC115568693
Glutathione peroxidase 2 (*gpx2*)	Ch_aur064.1	103–202	TGTGGCCTCGCTCTGAGGCACCACCACCCGGGACTACAGCGAGCTCAACCAGCTGCAGAGCAAGTACCCGCATCGGCTGGTGGTCCTGGGTTTTCCCTGT	gpx2
Hypoxanthine-guanine phosphoribosyltransferase (*hprt1*)	Ch_aur031.1	253–352	CTGAACAGGAACAGTGACCGCTCCATCCCAATGACAGTGGACTTCATCCGCCTCAAGAGCTACTGTAACGACCAGTCGACAGGTGAAATCAAAGTGATTG	hprt1
Interferon-induced GTP-binding protein Mx-like B (*mx-like*)	Ch_aur051.1	982–1081	CCATCTGATGCAGCTGAGAGAGTCGTCTTCCTCATTGATAAAGTGACAGCTTTCACTCAGGATGCCATCAGTCTGACCACAGGAGAAGAACTCAATTGTG	LOC115583120
Interleukin-1 β-like (*il1b-like*)	Ch_aur029.1	412–511	CCTACACCCAGTGCTGAGGCCGTAACTGTGACTCTGTGCATCAAGGACACAAATCTTTACCTGTCTTGTCACAAGGAAGGTGACGAGCCAACCTTGCATC	LOC115581181
Interleukin-6 isoform X1 (*il6*)	Ch_aur045.1	127–226	GTGATGCTGGCCGCTCTGCTTCAGTGTGCTCCCGGGGCTCCGATTGATGGCGCGCTCACTGACAATCCAGCAGGTGACACCTCAGGTGAAGAGTGGGAGA	LOC115579128
Interleukin-10-like (*il10-like*)	Ch_aur062.1	528–627	AGGTCTATACAAGGCCATGGGAGAGCTGGATCTGCTGTTCAACTACATTGAGACATATCTGGCTTCCAAACGGCACGGAACACATGTGGCCTCCGCTTGA	LOC115582730
Interleukin-12 subunit β-like (*il12b-like*)	Ch_aur032.1	388–487	GCACCTAACTATTCAGGCTCCTTCAAATGCACCTGGGCTAAAGCAGAGCACAGATCCAACGCCGCCGTGCTCCTGGTGAAGGCCGAACGTCATTTGGAGA	LOC115593944
Interleukin-17D (*il17d*)	Ch_aur010.1	415–514	CGCAGCACTCCGGTCTACGCTCCGTCTGTCATCCTGAGGAGAACCGGCTCCTGCCTCGGCGGCCGACACTCATACACAGAGATCTACGTCTCCATCGCGG	il17d
Interleukin-34 isoform X1 A (*il34*)	Ch_aur009.1	142–241	CGGTACATGAGGCACAACTTCCCCATCAAGTACACCATCAGGGTTCATCACAACGAAGTCTTTAAACTGTCAAACATCAGCAGACTGAGGTTACAGGTGG	il34
Macrophage colony-stimulating factor 1 receptor (*csf1r*)	Ch_aur060.1	2343–2442	CAAAAATTGTATTCACAGAGACATCGCTGCAAGGAATGTCCTGTTGACTGATCACAGAGTGGCCAAGATTTGTGACTTTGGTCTGGCACGTGACATCATG	csf1r
Mucin-2-like isoform X1 (*muc2-like*)	Ch_aur020.1	1740–1839	CTGTTCCCTCAGTGTGGAAAATGAGAATTACGCCAAACACTGGTGTGCCTTGCTGCTAAGTCCAGACAGCTCCTTTGCACAGTGCCGTTCAGCGGTGGAT	LOC115586438
Nuclear factor NF-kappa-B p100 subunit isoform X1 (*nfkb2*)	Ch_aur030.1	717–816	GGAGGCGTTCGGAGACTTTTCACCAACCGACGTTCACAAACAGTACGCCATTGTGTTCAAAACGCCGCCCTATCACAGCGCAGAGATCGAGCGGCCCGTC	nfkb2
Stromal cell-derived factor 1 (*cxcl12*)	Ch_aur016.1	197–296	AGAACAACAGGGAAGTTTGCATCAACCCGGAGACCAAGTGGCTGCAGCAGTACTTAAAGAACGCCATTAACAAGGTGAAGAAAAACCGAAGACGCAATAA	cxcl12
T-cell surface glycoprotein CD8 α chain-like isoform X3 (*cd8a-like*)	Ch_aur017.1	151–250	TGGTTTCGAGTGCTGGACAAATCTGGCATGGAATTCATTGGGTCTTTCAGCAATACTGGCGTGAAAAAACCAAATACAAAGCCTCCAACTTCCATCTTCA	LOC115583354
Toll-like receptor 2 isoform X1 (*tlr2*)	Ch_aur048.1	602–701	CGAGGTATGAGTCCGGTACTCTGGCATACGTTTGGCCGTTGGGTCGTGTCACTTTGAGCCTCCACAGTCCATTTTTAACAAATGAGGCCTTAGCCTCAGC	LOC115590525
Toll-like receptor 3 (*tlr3*)	Ch_aur014.1	799–898	AGCCAAGCTGATGGCAGCTTTCAGCCGTACAGCGCGGTGCTGCAGACCACTGAATCACTCAAAGTACTTCAGCTGCAATTCATGAAGGTGTTGATAGAAA	LOC115590587
Toll-like receptor 5 (*tlr5*)	Ch_aur053.1	1203–1302	CTTCCCTGCGTCTCTACCCAGATTAGATTATCTCCTGTTGAACGACAACAAGTTGACCTCCTCGTCAGTATACAGTCTCACACGGTTTGCCGATAATGCC	LOC115574263

**Table 4 genes-15-01390-t004:** Mean gene counts for snapper tissues with standard errors (SEs).

Treatment	Tissue	Statistic	Heat-Sensitive	Heat-Tolerant	Cold-Sensitive	Cold-Tolerant	Control
Actin cytoplasmic 1 (*actb*)	Fin clip	Mean	6464.6	6658.2	7090.1	6947.8	8314.7
		SE	170.6	153.6	488.1	323.8	370.8
	Liver	Mean	5317.4	5746.2	5364.2	6354.1	6790.7
		SE	418.9	622.1	347.9	286.7	302.0
	Head Kidney	Mean	23,691.5 *	26,749.1 *	15,841.8	15,277.8	17,274.3
		SE	1157.2	1392.0	748.0	941.8	776.9
Catalase isoform X2 (*cat*)	Fin clip	Mean	105.7 *	126.2 *	52.0	53.0	43.1
		SE	7.9	6.9	3.5	4.8	3.2
	Liver	Mean	1613.9	1540.7	1920.4	1898.8	2303.1
		SE	190.1	287.4	94.8	127.9	101.0
	Head Kidney	Mean	198.7 *	172.9 *	334.6	285.2	407.1
		SE	13.0	13.9	70.4	15.1	34.7
Cell surface glycoprotein MUC18-like isoform X (*muc18*-like)	Fin clip	Mean	87.6	123.3	100.7	104.6	123.3
		SE	5.5	3.5	7.7	6.8	8.2
	Liver	Mean	140.5 *	174.7	184.8	166.3 *	269.3
		SE	11.8	14.0	22.4	13.0	8.8
	Head Kidney	Mean	138.3 *	135.2 *	251.9	179.4	208.5
		SE	20.0	17.6	27.0	15.1	14.3
Complement C3-like A (*c3*-like)	Fin clip	Mean	25.3 *	32.8 *	9.1	12.4	27.7
		SE	2.3	5.5	3.4	4.0	25.7
	Liver	Mean	31,279.6	39,289.6	30,140.9	31,197.2	28,874.7
		SE	3841.9	5094.6	2518.0	3730.3	1613.4
	Head Kidney	Mean	0.0	0.0	2.8	0.0	0.0
		SE	0.0	0.0	3.0	0.0	0.0
Complement component C8 α chain (*c8a*)	Fin clip	Mean	0.0	1.4	0.0	0.0	3.1
		SE	0.0	1.5	0.0	0.0	3.3
	Liver	Mean	4625.5	5178.9	3922.7	3692.0	3908.2
		SE	333.1	567.4	329.8	288.1	142.5
	Head Kidney	Mean	0.0	0.0	1.6	0.0	0.0
		SE	0.0	0.0	1.7	0.0	0.0
Cryptochrome-1-like (*cry1-like*)	Fin clip	Mean	140.6 *	112.5 *	552.8 *	541.0 *	309.1
		SE	11.6	5.7	30.2	33.6	16.0
	Liver	Mean	58.7 *	45.9 *	314.1 *	332.2 *	155.3
		SE	6.8	4.9	22.7	26.4	13.4
	Head Kidney	Mean	127.7 *	113.7 *	341.3 *	339.4 *	212.9
		SE	19.1	6.1	35.1	9.4	21.5
Glutathione reductase mitochondrial isoform X2 (*gsr*)	Fin clip	Mean	145.6	135.1	151.2	144.0	141.3
		SE	13.2	15.5	9.3	6.7	11.1
	Liver	Mean	42.5 *	62.2 *^	96.5	78.6	97.8
		SE	3.2	3.5	5.6	5.3	9.1
	Head Kidney	Mean	208.4	268.1	339.3	315.4	276.8
		SE	22.2	24.7	28.4	26.8	9.6
Glutathione S-transferase A (*gsta*)	Fin clip	Mean	183.1	234.4	177.8	223.9	298.8
		SE	17.1	28.1	8.4	22.0	36.5
	Liver	Mean	333.7 *	203.4 *	91.2 *	94.9 *	981.8
		SE	49.1	40.0	13.9	13.4	305.0
	Head Kidney	Mean	319.0	307.9	744.8	726.1	358.6
		SE	61.7	45.5	218.5	98.7	50.6
Glyceraldehyde-3-phosphate dehydrogenase (*gapdh*)	Fin clip	Mean	5.3	23.7	8.1	5.6	19.1
		SE	2.3	10.6	3.1	3.1	15.1
	Liver	Mean	11,358.5	11,485.2	13,960.0	15,706.6	15,223.7
		SE	1038.7	1409.8	992.8	808.6	540.0
	Head Kidney	Mean	373.6	412.7	1838.8	970.9	549.2
		SE	118.4	115.1	725.8	169.6	197.0
Heat shock cognate 70 kDa protein (*hsp70*)	Fin clip	Mean	20,124.5 *	14,665.3 ^	15,513.6 *	14,063.7	12,488.3
		SE	1141.4	663.3	644.0	713.2	534.4
	Liver	Mean	15,171.0 *	14,161.8 *	8634.4	8257.5	8362.4
		SE	475.5	567.4	473.0	216.2	201.2
	Head Kidney	Mean	19,372.6 *	21,023.6 *	13,975.5	12,060.9	12,838.8
		SE	796.3	421.5	702.9	398.4	251.7
heat shock protein HSP 90-β isoform X2 (*hsp90ab1*)	Fin clip	Mean	16,357.6 *	13,753.0 *	10,585.8	8975.5	8616.6
		SE	829.6	637.8	533.4	304.0	268.4
	Liver	Mean	12,613.2 *	13,524.0 *	7554.1	7181.3	7100.6
		SE	488.0	479.3	289.5	190.9	184.9
	Head Kidney	Mean	14,686.9 *	16,384.4 *	8300.0	7199.7	7412.1
		SE	881.8	333.4	795.8	218.1	127.8
Hepcidin (*hamp*)	Fin clip	Mean	12.3	12.7	4.8	3.7	3.7
		SE	5.1	4.7	2.7	2.7	3.9
	Liver	Mean	11,855.0	7135.3	3733.8	4769.3	9498.1
		SE	3459.1	1325.1	1030.9	1463.4	1690.7
	Head Kidney	Mean	170.0 *	147.2 *	15.5	0.0	8.8
		SE	47.8	23.8	8.2	0.0	3.6
Insulin-like growth factor I isoform X1 (*igf1*)	Fin clip	Mean	7.7 *	16.6	20.6	22.2	72.1
		SE	3.6	3.9	4.5	2.0	7.0
	Liver	Mean	221.3 *	226.0 *	1091.4	957.7	1841.0
		SE	37.9	61.7	105.1	58.8	96.8
	Head Kidney	Mean	0.0 *	0.0 *	8.4 *	11.4 *	31.2
		SE	0.0	0.0	3.7	5.4	7.6
Insulin-like growth factor II (*igf2*)	Fin clip	Mean	69.7	93.5	70.0	94.7	87.1
		SE	9.6	10.1	5.0	7.2	3.7
	Liver	Mean	285.7 *	213.1 *^	154.7 *	228.9 *	1310.8
		SE	60.9	85.6	18.3	33.3	222.5
	Head Kidney	Mean	34.0	34.3	43.7	33.8	46.3
		SE	5.0	5.7	7.6	3.3	6.5
Mitochondrial uncoupling protein 2-like (*ucp2-like*)	Fin clip	Mean	8.0	2.9	0.0	2.3	3.5
		SE	2.3	2.0	0.0	1.6	3.7
	Liver	Mean	1161.5	1011.2	4767.2	6309.5	1333.0
		SE	255.4	313.2	749.8	444.2	128.6
	Head Kidney	Mean	4.0	0.0	29.6	22.5	8.0
		SE	2.2	0.0	16.0	6.3	4.0
Nuclear factor erythroid 2-related factor 2 (*nrf2*)	Fin clip	Mean	124.3 *	116.6 *	53.5	60.5	71.1
		SE	15.1	7.4	5.8	3.2	7.1
	Liver	Mean	330.4	228.5	333.7	316.1	262.9
		SE	48.3	23.1	29.8	19.8	27.1
	Head Kidney	Mean	307.3	322.9	245.9	309.4	338.5
		SE	11.0	16.0	10.4	14.8	19.6
Peroxiredoxin-1 (*prdx1*)	Fin clip	Mean	273.2 *	402.9 *	172.4	188.6	159.3
		SE	16.6	48.6	14.4	14.3	15.4
	Liver	Mean	688.3 *	624.2 *	750.7 *	777.2 *	1450.7
		SE	62.5	57.1	72.7	74.0	256.2
	Head Kidney	Mean	576.3	740.2 *	551.4	549.1	371.3
		SE	59.1	45.2	42.7	69.1	23.6
Peroxiredoxin-1-like (*prdx-like*)	Fin clip	Mean	327.3	316.5	323.9	308.3	336.4
		SE	13.2	11.7	30.6	25.3	22.1
	Liver	Mean	444.0 *	380.5 *	577.7	659.9	752.5
		SE	55.5	68.1	39.1	39.4	43.1
	Head Kidney	Mean	939.2	1310.6	876.1 *	816.1 *	1503.5
		SE	63.0	127.4	82.9	82.6	158.0
Peroxiredoxin-5 mitochondrial (*prdx5*)	Fin clip	Mean	242.6	315.5	569.0	517.8	383.6
		SE	12.0	61.8	39.7	21.8	15.7
	Liver	Mean	16.1 *	12.8 *	96.3	92.6	96.8
		SE	2.6	3.8	7.4	7.1	5.8
	Head Kidney	Mean	72.0	56.7	146.4	128.6	121.4
		SE	15.1	5.7	16.5	13.3	10.4
Prostaglandin G/H synthase 2 (*ptgs2*)	Fin clip	Mean	346.8	171.4	165.4	125.1	166.5
		SE	70.1	49.4	35.6	12.2	21.0
	Liver	Mean	22.7 *	79.2 *^	6.4	0.0	1.2
		SE	9.0	7.3	4.8	0.0	1.3
	Head Kidney	Mean	1.9	7.1	1.2	1.4	0.0
		SE	2.0	2.5	1.3	1.5	0.0
Serotransferrin-like (*tf-like*)	Fin clip	Mean	53.7	68.7	32.7	36.6	56.1
		SE	5.6	12.4	3.4	8.8	33.0
	Liver	Mean	80,683.6	82,194.8	61,926.1	56,895.0	81,079.4
		SE	6054.3	9636.5	5109.7	7651.3	4822.1
	Head Kidney	Mean	1.8	0.0	0.0	1.2	0.0
		SE	1.9	0.0	0.0	1.3	0.0
Superoxide dismutase [Cu-Zn] (*sod1*)	Fin clip	Mean	389.1	313.2	471.0	383.6	342.3
		SE	34.8	17.4	30.8	24.8	13.3
	Liver	Mean	426.6 *	311.2 *	710.3 *	746.8 *	1177.7
		SE	49.4	45.2	33.7	35.7	66.8
	Head Kidney	Mean	332.5 *	361.0	475.6	411.1	521.9
		SE	18.5	16.3	35.4	23.5	31.2
Suppressor of cytokine signalling 3 (*socs3*)	Fin clip	Mean	281.2	243.7	209.8	190.7	466.9
		SE	38.8	39.8	33.1	15.6	58.3
	Liver	Mean	738.1 *	690.1 *	38.6	65.8	48.5
		SE	319.6	291.2	5.1	11.8	13.2
	Head Kidney	Mean	1126.1	1231.6	204.7	147.5	167.2
		SE	394.1	291.6	44.9	13.4	16.8
Thioredoxin-dependent peroxide reductase mitochondrial (*prdx3*)	Fin clip	Mean	189.9 *	173.7 *	198.6 *	232.3 *	103.2
		SE	21.4	20.3	19.8	18.2	6.9
	Liver	Mean	195.6 *	202.8 *	299.1	276.5	400.7
		SE	14.3	21.0	18.6	12.4	32.5
	Head Kidney	Mean	163.0	209.0	202.3	186.0	220.0
		SE	11.9	13.4	9.6	13.4	11.9
Transforming growth factor β-1 proprotein-like isoform X1 (*tgfb1-like*)	Fin clip	Mean	321.0	300.7	361.6	371.9	331.3
		SE	22.0	57.5	16.7	18.9	25.0
	Liver	Mean	41.3	32.4	36.8	32.7	48.4
		SE	5.8	2.8	6.2	2.5	4.7
	Head Kidney	Mean	249.0	269.2	228.5	185.9	190.0
		SE	14.9	13.5	8.6	13.5	11.0

* Mean gene counts (*n* = 10) are shown with standard errors for heat-sensitive, heat-tolerant, cold-sensitive, cold-tolerant and ambient (control) fish. A significant difference using Fisher’s least significant difference test (*p* = 0.00033 = 0.05/(6*25)) is indicated by * for differences compared with the control, whilst ^ indicates a difference between the heat-sensitive and heat-tolerant groups.

## Data Availability

The data generated and analysed in this study are available upon request from the corresponding author. Access to these data is contingent upon obtaining appropriate consent from the guardians (kaitiaki) of snapper, the Māori indigenous people of New Zealand, to honour their role as stewards of this taonga (treasured species). Potential requestors must provide a detailed explanation of their intended research purpose, including the potential outcomes and benefits of their work. Additionally, requestors are expected to outline any benefit-sharing arrangements before data access can be requested. This ensures that the use and reuse of the data respects the cultural values and guardianship of Māori over natural resources.

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
