# Peer review of "Development of a Novel Stress and Immune Gene Panel for the Australasian Snapper (*Chrysophrys auratus*)"

_genes, 2024, doi:10.3390/genes15111390_

Round 1

Reviewer 1 Report

Comments and Suggestions for Authors

Given that the Australasian snapper is a valuable aquaculture species, information regarding its welfare (including stress, which is an important aspect of proper husbandry) is crucial from the perspective of fish farmers. Additionally, in the context of globally rising temperatures, the research conducted by the Authors may prove to be significant for the biology of the species, which, unfortunately, will have to adapt to climate change.

However, I miss some key information in the paper, such as:

1.       Why did the Authors choose these particular temperatures for the experiment? What is the standard rearing temperature for this species under aquaculture conditions? 

2.       Could the Authors provide basic zootechnical information, such as the average weight and length of the fish? 

3.       I am missing information on fish mortality in the different temperature variants, which would also provide valuable insights. For instance, if all the fish died at 31°C or 7°C, I don't see much sense in looking for stress markers under such conditions.

Also I have one comment to take under the considerations by the Authors:

The rules for writing gene names vary depending on the organism, but in molecular biology, there is a general convention that dictates how we write genes across different species.

For humans: Genes are usually written in uppercase italics (e.g., SOD). 

For animals other than humans, including fish: Gene names are typically written in lowercase italics (e.g., sod for fish, mice, or other model animals). Proteins, however, are written in uppercase (e.g., SOD) (See for example The Zebrafish Model Organism Database or HUGO Gene Nomenclature Committee).

Author Response

Thank you for the review. Changes to the manuscript are shown in red/blue. Additionally, to the edits described below we have made some minor edits which are also highlighted throughout the manuscript document.

Comment 1: Why did the Authors choose these particular temperatures for the experiment? What is the standard rearing temperature for this species under aquaculture conditions? 

Response 1:

Thank you for your comment. There is no standard rearing temperature per se as there is currently no commercial farming. For improved growth larvae are heated up to 22 degrees [1] or 22 -24 degrees [2]. For this study larvae were kept at ambient temperature with no additional heating: 19-21 degrees (with heaters set to 20 degrees, if ambient temperatures would go lower). Fish were then kept at 18 degrees for several weeks as acclimation time until the experiment started. 

References

  1. Samuels, G.; Hegarty, L.; Fantham, W.; Ashton, D.; Blommaert, J.; Wylie, M.J.; Moran, D.; Wellenreuther, M. Generational breeding gains in a new species for aquaculture, the Australasian snapper. Aquaculture 2024, 586, doi:ARTN 740782

10.1016/j.aquaculture.2024.740782.

  1. Moran, D.; Schleyken, J.; Flammensbeck, C.; Fantham, W.; Ashton, D.; Wellenreuther, M. Enhanced survival and growth in the selectively bred (Australasian snapper, ta over line mure). Aquaculture 2023, 563, doi:ARTN 73897010.1016/j.aquaculture.2022.738970.

These temperature settings are all aligned with natural conditions near Nelson, see Figure 1.

We have added extra information to the manuscript in the introduction section (Lines 73-86) to explain standard temperatures that snapper are exposed to and why the temperatures in the experiment were selected.  “Snapper has a wide distribution across New Zealand and Australia, inhabiting nearly all inshore environments down to depths of 200 m. Snapper has a wide distribution across New Zealand and Australia, inhabiting nearly all inshore environments down to depths of 200 m [3,4]. In New Zealand, Snapper is predominantly found in the northern half of the North Island and the northern part of the South Island [4], occasionally occurring in the southern South Island [5], where temperatures can go well below 10 ºC. Like many other Sparidae species, they experience a broad range of natural temperatures [6-9]. For instance, their range extends as far north as Mackay, where sea surface temperatures (SSTs) can reach up to 30°C [10]. Moreover, experimental work has repeatedly shown that snapper can adjust effectively to both warm and cold temperature treatments, exhibiting minimal mortality even after months of exposure [8,11]. Herein, we aimed to exceed the species’ thermal tolerance limits by exposing snapper to extreme temperatures outside of the seasonal temperature range, since climate change may expose snapper to these extremes in the future.”

References (please note reference numbers differ in the manuscript)

  1. Leach, F. Fishing in Pre-European New Zealand. NZ ASA 2007, 15, 469-471.
  2. Parsons, D.; Sim-Smith, C.; Cryer, M.; Francis, M.; Hartill, B.; Jones, E.; Le Port, A.; Lowe, M.; McKenzie, J.; Morrison, M. Snapper (Chrysophrys auratus): a review of life history and key vulnerabilities in New Zealand. New Zeal J Mar Fresh 2014, 48, 256-283.
  3. Graham, D. A treasury of New Zealand fish. Wellington (New Zealand): AH & AW Reed 1953.
  4. Francis, M.P. Coastal fishes of New Zealand. An identification guide. 3rd edn. ed.; Reed Books: Auckland, 2001; Vol. 3rd edn.
  5. Francis, M.P. Geographic distribution of marine reef fishes in the New Zealand region. New Zeal J Mar Fresh 1996, 30, 30-55.
  6. Wellenreuther, M.; Le Luyer, J.; Cook, D.; Ritchie, P.A.; Bernatchez, L. Domestication and temperature modulate gene expression signatures and growth in the Australasian snapper Chrysophrys auratus. G3: Genes, Genomes, Genetics 2019, 9, 105-116.
  7. Moran, D.; Schleyken, J.; Flammensbeck, C.; Fantham, W.; Ashton, D.; Wellenreuther, M. Enhanced survival and growth in the selectively bred Chrysophrys auratus (Australasian snapper, tāmure). Aquaculture 2023, 563 (1)
  8. Ferrell, D. Assessment of the fishery for snapper (Pagrus auratus) in Queensland and New South Wales. Final Report, FRDC Project 93/074 1993, 143.
  9. Bowering, L.R.; McArley, T.J.; Devaux, J.B.; Hickey, A.J.; Herbert, N.A. Metabolic resilience of the Australasian snapper (Chrysophrys auratus) to marine heatwaves and hypoxia. Frontiers in Physiology 2023, 14, 1215442.
  10. Blommaert, J.; Sandoval-Castillo, J.; Beheregaray, L.; Wellenreuther, M. Peering into the gaps: Long-read sequencing illuminates structural variants and genomic evolution in the Australasian snapper. Genomics 2024, 110929, doi:https://doi.org/10.1016/j.ygeno.2024.110929.
  11. Teletchea, F.; Fontaine, P. Levels of domestication in fish: implications for the sustainable future of aquaculture. Fish Fish. 2014, 15, 181-195.
  12. Mignon-Grasteau, S.; Boissy, A.; Bouix, J.; Faure, J.-M.; Fisher, A.D.; Hinch, G.N.; Jensen, P.; Le Neindre, P.; Mormède, P.; Prunet, P. Genetics of adaptation and domestication in livestock. Livestock Production Science 2005, 93, 3-14.
  13. Chiswell, S.M.; Sutton, P.J. Relationships between long-term ocean warming, marine heat waves and primary production in the New Zealand region. New Zeal J Mar Fresh 2020, 54, 614-635.
  14. Law, C.S.; Rickard, G.J.; Mikaloff-Fletcher, S.E.; Pinkerton, M.H.; Behrens, E.; Chiswell, S.M.; Currie, K. Climate change projections for the surface ocean around New Zealand. New Zeal J Mar Fresh 2018, 52, 309-335, doi:10.1080/00288330.2017.1390772.
  15. Environment, M.f.t. Likely climate change impacts in New Zealand Available online: https://www.mfe.govt.nz/climate-change/likely-impacts-of-climate-change/likely-climate-change-impacts-nz (Accessed on: January 2021).

Comment 2: Could the Authors provide basic zootechnical information, such as the average weight and length of the fish? 

Response 2:

Thank you for your comment. We have included this information in the experimental design section (lines 157-164) and added a table with the information on average weight and length of snapper from each treatment group.

The new text reads: “Head kidney, liver and a fin sample was taken from a total of 50 individuals representing control (n=10), temperature sensitive (n=20, 10 per treatment group) and temperature tolerant individuals (n=20, 10 per treatment group) (see Table 1 for phenotypic data on the different sample groups). The experimental design is shown in Figure 1.”

The new table is shown below:

Table 1. Length (mm, mean ± SE) and weight (g, mean ± SE) measurements of 50 juvenile snapper exposed to temperature challenge and sampled for gene expression analysis.

Treatment group

Sample size (n)

Fork length [mm]

Weight [g]

Control

10

111.7 ± 3

29.8 ± 3 

Cooling: Temperature sensitive

10

107.1 ± 4

26.2 ± 3 

Cooling: Temperature tolerant

10

116.0 ± 4

33.7 ± 4 

Heating: Temperature sensitive

10

108.1 ± 3

25.1 ± 2 

Heating: Temperature tolerant

10

113.6 ± 3

30.9 ± 2 

Comment 3: I am missing information on fish mortality in the different temperature variants, which would also provide valuable insights. For instance, if all the fish died at 31°C or 7°C, I don't see much sense in looking for stress markers under such conditions.

Response 3:

Thank you for your comment. No fish died in the treatment groups as they were removed when they showed signed of distress and then sampled for this study.

However, during the rearing of fish in tanks during acclimation time (a period of approximately six weeks), ten individuals from the heating treatment pool and two individuals from the cooling treatment pool died. Moreover, during the heating phase of the heating treatment two more mortalities occurred, but none of these fish were sampled for this study. A larger experiment was conducted for additional study outcomes that will be described in future publications.

For this study, 10 fish were selected for the control group, then the first ten fish, which were the most sensitive ones to extreme temperature changes. At the end of the experiment, the ten most temperature tolerant individuals were selected. If the decrease and increase of temperature, respectively, had been slowly reversed, most fish would have been able to recover and maintain homeostasis. The details of the full experiment design are described in section 2.1 (lines 122-166)

Comment 4: Also I have one comment to take under the considerations by the Authors:

The rules for writing gene names vary depending on the organism, but in molecular biology, there is a general convention that dictates how we write genes across different species.

For humans: Genes are usually written in uppercase italics (e.g., SOD). 

For animals other than humans, including fish: Gene names are typically written in lowercase italics (e.g., sod for fish, mice, or other model animals). Proteins, however, are written in uppercase (e.g., SOD) (See for example The Zebrafish Model Organism Database or HUGO Gene Nomenclature Committee).

Response 4: Thank you for pointing this out. We have edited the manuscript through-out including tables 2, 3 and 4 and Figures 2, 3, and 4 to change all gene name abbreviations to lowercase and italics and kept protein names as uppercase.

Reviewer 2 Report

Comments and Suggestions for Authors

The article is well structured, cohesive and very interesting. The authors provide clear results, perfectly in line with their discussions and conclusions. The materials and methods are well explained and of fairly immediate comprehension. The authors could emphasize the novelty of their research more in the introduction. Also, adding a few critical notes about any limitations of the study in the discussions would be optimal. This would further improve the quality of the manuscript. Some of the data in the bibliography are quite distant; if the authors could, where possible, provide more up-to-date data, it would be desirable. Overall, I suggest a minor revision.

Author Response

Dear Reviewer:

Thank you for the review. Changes to the manuscript are shown in red/blue. Additionally, to the edits described below we have made some minor edits which are also highlighted throughout the manuscript document.

Comment 1:  The authors could emphasize the novelty of their research more in the introduction.

Response 1: Thank you for your comment. The following text has been added to the manuscript (Lines 102-109) “In this study, we aim to identify and develop novel biomarkers associated with the thermotolerance of Australasian snapper using NanoString Technologies, Inc. This is the first time such an approach has been applied to this species, making it a novel contribution to the field. By focusing on genes responsive to different temperature settings, we will enhance our understanding of the physiological and molecular mechanisms underpinning the snapper's resilience to thermal stress. This focus on biomarkers will facilitate the selection of breeding stock with superior thermal tolerance, contributing to a deeper understanding of the snapper’s adaptive capacity in fluctuating environments.”

Comment 2: Also, adding a few critical notes about any limitations of the study in the discussions would be optimal. This would further improve the quality of the manuscript.

Response 2: Thank you for your comment. There is a section in the discussion describing the limitations of this study (lines 374-385). This reads “The limitations of the study are that annotations for the snapper and gilthead sea-bream genome are still limited, so the results must be interpreted with some caution. Work is currently underway to use an improved genome assembly for snapper [73] to add im-proved gene annotations, and this may support future effort to develop an improved pan-el. Also, genes with -like in their name have been discussed as if they were the gene. Addi-tionally, the study uses captive fish which may differ compared with wild fish, and this could mean that stress responses are limited or exacerbated [74,75]. Identifying genetic traits that increase tolerance to temperature changes carries significant applied value as this knowledge can be used to selectively breed snapper for these traits to improve resili-ence to temperature fluctuations and ultimately to make this a more commercially viable aquaculture species, something that is urgently needed in New Zealand due to increasing impacts of climate change on the aquaculture sector [76-78].”

Comment 3: Some of the data in the bibliography are quite distant; if the authors could, where possible, provide more up-to-date data, it would be desirable.

Response 3: Thank you for pointing this out. We have reviewed the manuscript and could not work out which papers you maybe referring too. We have included some additional references to the manuscript in new paragraphs that have been added that may help to explain why such work has been referenced. Some of the relevant references are very old. In addition to text included in response 1. We have added extra information to the manuscript in the introduction section (Lines 73-76) to explain standard temperatures that snapper are exposed to and why the temperatures in the experiment were selected. “Snapper has a wide distribution across New Zealand and Australia, inhabiting nearly all inshore environments down to depths of 200 m. Snapper has a wide distribution across New Zealand and Australia, inhabiting nearly all inshore environments down to depths of 200 m [3,4]. In New Zealand, Snapper is predominantly found in the northern half of the North Island and the northern part of the South Island [4], occasionally occurring in the southern South Island [5], where temperatures can go well below 10 ºC. Like many other Sparidae species, they experience a broad range of natural temperatures [6-9]. For instance, their range extends as far north as Mackay, where sea surface temperatures (SSTs) can reach up to 30°C [10]. Moreover, experimental work has repeatedly shown that snapper can adjust effectively to both warm and cold temperature treatments, exhibiting minimal mortality even after months of exposure [8,11]. Herein, we aimed to exceed the species’ thermal tolerance limits by exposing snapper to extreme temperatures outside of the seasonal temperature range, since climate change may expose snapper to these extremes in the future.”

References (please note reference numbers differ in the manuscript)

  1. Leach, F. Fishing in Pre-European New Zealand. NZ ASA 2007, 15, 469-471.
  2. Parsons, D.; Sim-Smith, C.; Cryer, M.; Francis, M.; Hartill, B.; Jones, E.; Le Port, A.; Lowe, M.; McKenzie, J.; Morrison, M. Snapper (Chrysophrys auratus): a review of life history and key vulnerabilities in New Zealand. New Zeal J Mar Fresh 2014, 48, 256-283.
  3. Graham, D. A treasury of New Zealand fish. Wellington (New Zealand): AH & AW Reed 1953.
  4. Francis, M.P. Coastal fishes of New Zealand. An identification guide. 3rd edn. ed.; Reed Books: Auckland, 2001; Vol. 3rd edn.
  5. Francis, M.P. Geographic distribution of marine reef fishes in the New Zealand region. New Zeal J Mar Fresh 1996, 30, 30-55.
  6. Wellenreuther, M.; Le Luyer, J.; Cook, D.; Ritchie, P.A.; Bernatchez, L. Domestication and temperature modulate gene expression signatures and growth in the Australasian snapper Chrysophrys auratus. G3: Genes, Genomes, Genetics 2019, 9, 105-116.
  7. Moran, D.; Schleyken, J.; Flammensbeck, C.; Fantham, W.; Ashton, D.; Wellenreuther, M. Enhanced survival and growth in the selectively bred Chrysophrys auratus (Australasian snapper, tāmure). Aquaculture 2023, 563 (1)
  8. Ferrell, D. Assessment of the fishery for snapper (Pagrus auratus) in Queensland and New South Wales. Final Report, FRDC Project 93/074 1993, 143.
  9. Bowering, L.R.; McArley, T.J.; Devaux, J.B.; Hickey, A.J.; Herbert, N.A. Metabolic resilience of the Australasian snapper (Chrysophrys auratus) to marine heatwaves and hypoxia. Frontiers in Physiology 2023, 14, 1215442.
  10. Blommaert, J.; Sandoval-Castillo, J.; Beheregaray, L.; Wellenreuther, M. Peering into the gaps: Long-read sequencing illuminates structural variants and genomic evolution in the Australasian snapper. Genomics 2024, 110929, doi:https://doi.org/10.1016/j.ygeno.2024.110929.
  11. Teletchea, F.; Fontaine, P. Levels of domestication in fish: implications for the sustainable future of aquaculture. Fish Fish. 2014, 15, 181-195.
  12. Mignon-Grasteau, S.; Boissy, A.; Bouix, J.; Faure, J.-M.; Fisher, A.D.; Hinch, G.N.; Jensen, P.; Le Neindre, P.; Mormède, P.; Prunet, P. Genetics of adaptation and domestication in livestock. Livestock Production Science 2005, 93, 3-14.
  13. Chiswell, S.M.; Sutton, P.J. Relationships between long-term ocean warming, marine heat waves and primary production in the New Zealand region. New Zeal J Mar Fresh 2020, 54, 614-635.
  14. Law, C.S.; Rickard, G.J.; Mikaloff-Fletcher, S.E.; Pinkerton, M.H.; Behrens, E.; Chiswell, S.M.; Currie, K. Climate change projections for the surface ocean around New Zealand. New Zeal J Mar Fresh 2018, 52, 309-335, doi:10.1080/00288330.2017.1390772.
  15. Environment, M.f.t. Likely climate change impacts in New Zealand Available online: https://www.mfe.govt.nz/climate-change/likely-impacts-of-climate-change/likely-climate-change-impacts-nz (Accessed on: January 2021).